# Massive, long-lived electrostatic potentials in a rotating mirror plasma

E. J. Kolmes [1] ✉, I. E. Ochs [1], J.-M. Rax [2,3] & N. J. Fisch [1]

Hot plasma is highly conductive in the direction parallel to a magnetic field. This often means that the electrical potential will be nearly constant along any given field line. When this is the case, the cross-field voltage drops in open-field-line magnetic confinement devices are limited by the tolerances of the solid materials wherever the field lines impinge on the plasma-facing components. To circumvent this voltage limitation, it is proposed to arrange large voltage drops in the interior of a device, but coexist with much smaller drops on the boundaries. To avoid prohibitively large dissipation requires both preventing substantial drift-flow shear within flux surfaces and preventing large parallel electric fields from driving large parallel currents. It is demonstrated here that both requirements can be met simultaneously, which opens up the possibility for magnetized plasma tolerating steady-state voltage drops far larger than what might be tolerated in material media.

The largest steady-state laboratory electrostatic potential in the world was likely produced by the Van de Graaf-like pelletron generator at the Holifield facility at Oak Ridge National Laboratory. Housed within a 30-meter-tall, 10-meter-diameter pressure chamber filled with insulating $SF_6$ gas, the generator was able to maintain electrostatic potentials of around 25 MV[1]. The main obstacle limiting the production of even greater potentials in the laboratory is the breakdown electric field of the surrounding medium.

A fully ionized plasma is a promising setting in which to pursue very large voltage drops, in part because it is by definition already broken down. Moreover, once a magnetic field is added, plasma has a very attractive property: charged particles cannot move across the magnetic field lines, as they are confined on helical paths along the field. As long as a stable plasma equilibrium is identified, the particles can only move across the field as a result of collisions and cross-field drifts, and thus are theoretically capable of coexisting with much larger electric fields than could a gas.

Unfortunately, this nice confinement property only works along two out of three of the spatial dimensions, with electrons free to stream along magnetic field lines, shorting out any "parallel" electric field. For instance, in a cylinder with the magnetic field pointing along the axis, the medium is highly insulating along the radial and azimuthal directions, but highly conductive along the axial direction. Thus, one

must either loop the fields around on themselves, which introduces a variety of instabilities and practical difficulties, or one must introduce a potential drop along the field lines.

This latter approach is closely related to a magnetic confinement concept known as the centrifugal mirror trap, which has applications both in nuclear fusion[2–9] and mass separation[2,10–13]. These devices typically consist of an approximately radial electric field superimposed on an approximately axial magnetic field, such that the resulting $\mathbf{E} \times \mathbf{B}$ drifts produce azimuthal rotation. By pinching the ends of the device to a smaller radius, particles must climb a centrifugal potential in order to exit the device and thus can be confined. The conventional strategy for imposing the desired electric field is to place nested ring electrodes at the ends of the device, relying on the high parallel conductivity to propagate the potential into the core. However, this strategy fundamentally limits the achievable core electric field, and thus the achievable centrifugal potential, since one must avoid arcing across the end electrodes. The question of confining the electric potential to the center of the device is thus not only of academic interest but also of significant practical interest in such centrifugal fusion concepts.

In this paper, we propose an alternative arrangement, in which the voltage drop is produced in the interior of the plasma using either wave-particle interactions or neutral beams[14–19]. Wave-particle interactions have been proposed to move ions across field lines for the

[1]Department of Astrophysical Sciences, Princeton University, Princeton, NJ 08544, USA. [2]Andlinger Center for Energy and the Environment, Princeton University, Princeton, NJ 08544, USA. [3]IJCLab, Université de Paris-Saclay, 91405 Orsay, France. ✉e-mail: ekolmes@princeton.edu

purpose of achieving the alpha channeling effect, where the main purpose is to remove ions while extracting their energy. Here the focus is instead on moving net charge across field lines. Moving charge across field lines could sustain a potential difference in the interior of the system that is higher than the potential across the plasma-facing material components at the ends.

In order for wave-driven electric fields to entirely circumvent the most important material restrictions on electrode-based systems, it is necessary that the voltage drop not only be driven in the interior of the plasma but that it be contained there. Otherwise, the induced voltage drops will simply incur power dissipation at the plasma boundaries no matter where along the magnetic surface the voltage drop is induced. In other words, there must be steady-state electric fields parallel to the magnetic field lines.

Relatively small parallel electric fields have long been predicted (and observed) in mirror-like configurations[20–27]. Larger fields have been predicted[2,28,29] and observed[9] for some systems, but have typically not been achievable in higher-temperature steady-state laboratory systems[2], for two very good reasons. First: if the flux surfaces are not close to being isopotential surfaces, then the rotation may be strongly sheared along a given flux surface. This would tend to lead to significant dissipation, and perhaps also to twisting-up of the magnetic field as the sheared plasma carries the field lines along with it. Second: large parallel fields typically incur large Joule heating. The resulting dissipation from either of these effects could be prohibitively large for many applications.

This paper addresses the following question: is it possible to eliminate these large dissipation terms while maintaining a large parallel component of **E**? This requires, firstly, revisiting conventional assumptions about *isorotation*: the conditions under which the plasma on each flux surface will rotate with a fixed angular velocity. While the absence of parallel electric fields is a sufficient condition for isorotation – this is Ferraro's isorotation law[30] – we will show that it is not a necessary condition. Moreover, there are cases in which large parallel fields can exist with vanishingly small parallel currents. In principle, then, it is possible to construct extremely low-dissipation systems with both (1) a very large voltage drop across the field lines in the interior of the plasma and (2) little or no voltage drop across the field lines at the edges of the plasma. Of course, being possible is not the same as being easy, and meeting all of these conditions simultaneously puts stringent conditions on the system.

However, if a contained voltage drop were attainable and stable, the resulting possibilities could be striking. Fast rotation is desirable for fusion technologies and mass filtration; moreover, the possibility of achieving ultra-high DC voltage drops in the laboratory – and, particularly, of decoupling the achievable voltages from the constraints associated with the material properties of solids – could be even more broadly useful.

## Results
### Shear
This section will describe the necessary and sufficient conditions for isorotation in an axisymmetric plasma. The usual isorotation picture[30,31], in which each flux surface is a surface of constant voltage, is one special case of these conditions.

Consider an axisymmetric plasma – that is, in $(r, \theta, z)$ cylindrical coordinates, suppose that the system is symmetric with respect to $\theta$. Suppose there is no $\theta$-directed magnetic field. Define the flux $\psi$ by

$$\psi \doteq \int_0^r r' B_z(r',z)\, dr'. \tag{1}$$

This definition, combined with the requirement that $\nabla \cdot \mathbf{B} = 0$, implies that

$$\mathbf{B} = -\frac{1}{r}\frac{\partial \psi}{\partial z}\,\hat{r} + \frac{1}{r}\frac{\partial \psi}{\partial r}\,\hat{z} = \nabla\psi \times \nabla\theta. \tag{2}$$

If the current **j** satisfies $\mathbf{j} \cdot \nabla\psi = 0$, it is possible to find a third coordinate $\chi$ and scalar function $\gamma$ such that[32,33]

$$\mathbf{B} = \nabla\chi + \gamma\nabla\psi. \tag{3}$$

Eqs. (2) and (3) imply that

$$\nabla\chi \cdot (\nabla\psi \times \nabla\theta) = B^2. \tag{4}$$

In the vacuum-field limit, we can take $\gamma \to 0$ and $\chi$ to be the magnetic scalar potential. This is possible because, in the absence of plasma currents, the curl of **B** vanishes everywhere in the interior of the plasma and the Helmholtz decomposition can be written in terms of a pure scalar potential.

Suppose the electric field **E** is given by $\mathbf{E} = -\nabla\phi$. Then the $\mathbf{E} \times \mathbf{B}$ drift is given by

$$\mathbf{v}_{E \times B} = -\frac{\nabla\phi \times \mathbf{B}}{B^2} \tag{5}$$

$$= -\frac{1}{B^2}\left(\frac{\partial \phi}{\partial \psi} - \gamma\frac{\partial \phi}{\partial \chi}\right)\nabla\psi \times \nabla\chi \tag{6}$$

and the $\mathbf{E} \times \mathbf{B}$ rotation frequency is

$$\Omega_E = \mathbf{v}_{E \times B} \cdot \nabla\theta = \frac{\partial \phi}{\partial \psi} - \gamma\frac{\partial \phi}{\partial \chi}. \tag{7}$$

Then, assuming a nonvanishing field,

$$\mathbf{B} \cdot \nabla\Omega_E = 0 \quad \text{iff.} \quad \frac{\partial}{\partial \chi}\left(\frac{\partial \phi}{\partial \psi} - \gamma\frac{\partial \phi}{\partial \chi}\right) = 0. \tag{8}$$

Eq. (8) is satisfied by any potential of the form $\phi = \phi_{\|} + \phi_{\perp}$, where $\mathbf{B} \cdot \nabla\phi_{\perp} = 0$ and $\mathbf{B} \times \nabla\phi_{\|} = 0$. In the vacuum-field case, the situation is particularly simple: $\mathbf{B} \cdot \nabla\Omega_E$ vanishes if and only if

$$\phi = \phi_{\|}(\chi) + \phi_{\perp}(\psi) \tag{9}$$

for arbitrary functions $\phi_{\|}$ and $\phi_{\perp}$. These two potentials will correspond to electric fields in the parallel and perpendicular directions, respectively. The entire system will rotate as a solid body if, in addition, $\phi_{\perp}$ is a linear function of $\psi$.

For some systems, the diamagnetic drift velocities may not be negligible compared with the $\mathbf{E} \times \mathbf{B}$ velocity. In that case, the viscous dissipation typically depends on shear in the combined drift velocity[34–37]. At least in the isothermal case, the generalization of Eq. (9) is straightforward. Define the effective (electrochemical) potential $\varphi_s$ for species $s$ is defined by

$$\varphi_s \doteq \phi - \frac{T_s}{q_s}\log n_s, \tag{10}$$

where $n_s$, $T_s$, and $q_s$ are the density, temperature, and charge of species $s$. This object is sometimes known as the "thermal" or "thermalized" potential in the Hall thruster literature[38–40]. In terms of $\varphi_s$, the combined

rotation frequency is

$$\Omega_{s,\text{tot}} = \frac{\partial \varphi_s}{\partial \psi} - \gamma \frac{\partial \varphi_s}{\partial \chi}, \tag{11}$$

with

$$\mathbf{B} \cdot \nabla \Omega_{s,\text{tot}} = 0 \quad \text{iff.} \quad \frac{\partial}{\partial \chi}\left(\frac{\partial \varphi_s}{\partial \psi} - \gamma \frac{\partial \varphi_s}{\partial \chi}\right) = 0, \tag{12}$$

reducing to the requirement that

$$\varphi_s = \varphi_{s,||}(\chi) + \varphi_{s,\perp}(\psi) \tag{13}$$

in the vacuum-field limit.

The classical form of the isorotation theorem takes the electrostatic potential to be a flux function: that is, $\phi = \phi(\psi)$. The extension to a generalized potential – that is, $\varphi_s = \varphi_s(\psi)$ – has been known for some time in the literature on plasma propulsion[38–40]. These previous cases provide sufficient conditions for isorotation. The more general expression derived here is the necessary and sufficient condition for isorotation.

In cases with very fast rotation – that is, with $\Omega_{s,\text{tot}}$ comparable to the particle's gyrofrequency – additional inertial effects can become relevant. The centrifugal $\mathbf{F} \times \mathbf{B}$ drift can be incorporated by including an appropriate term in the electrochemical potential; in cases where the centrifugal force $m_s r \Omega_{s,\text{tot}}^2 \hat{r}$ is the gradient of a centrifugal potential, this is as simple as adding that potential to $\varphi_s$. In the vacuum-field limit, the effective perpendicular potential

$$\varphi_{cs,\text{eff}} = -\int_0^\psi \frac{m_s \Omega_{s,\text{tot}}^2(\chi,\psi') B_z(\chi,\psi')\, d\psi'}{q_s B^2(\chi,\psi')}, \tag{14}$$

leads to the appropriate drift frequency $\partial \varphi_{cs,\text{eff}}/\partial \psi$, whether or not the centrifugal force has a curl. However, note that once $\Omega_{s,\text{tot}}$ is comparable to the gyrofrequency, excursions of the particle trajectories from the flux surfaces can also become significant.

## An example
Consider a magnetic field given[25,31,41] by $\mathbf{B} = \nabla \chi$, where

$$\chi = B_0 L \left[\frac{z}{L} - \frac{\alpha}{2\pi} \sin\left(\frac{2\pi z}{L}\right) I_0\left(\frac{2\pi r}{L}\right)\right]. \tag{15}$$

Here $I_\ell$ denotes a modified Bessel function of the first kind. This scalar potential leads to

$$B_z = B_0 \left[1 - \alpha \cos\left(\frac{2\pi z}{L}\right) I_0\left(\frac{2\pi r}{L}\right)\right] \tag{16}$$

and

$$B_r = -B_0 \alpha \sin\left(\frac{2\pi z}{L}\right) I_1\left(\frac{2\pi r}{L}\right). \tag{17}$$

Then the flux function can be written as

$$\psi = \frac{B_0 r^2}{2} \left[1 - \frac{\alpha L}{\pi r} \cos\left(\frac{2\pi z}{L}\right) I_1\left(\frac{2\pi r}{L}\right)\right]. \tag{18}$$

Having an explicit form for $\chi$ and $\psi$ makes it straightforward to construct an example in which the isopotential surfaces close and

$\mathbf{B} \cdot \nabla \Omega_E$ vanishes. Figure 1 shows one such example, with

$$\frac{\phi}{\phi_0} = -\frac{\psi(r,z)}{\psi(L/10,0)} - \left(\frac{\chi(r,z)}{\chi(0,L/2)}\right)^2. \tag{19}$$

## Parallel currents
The potential structure shown in Fig. 1 avoids parallel shear in $\Omega_E$. However, large parallel electric fields are likely to lead to large parallel currents. It might be possible to maintain such fields with means of a non-inductive current drive[42], but for small power dissipation, the noninductive current drive must be efficient, whereas the return current must encounter high plasma resistivity, which is unlikely in the hot plasmas considered here which would have large parallel conductivity.

In order to understand the behavior of these parallel currents, consider a simple two-fluid model for steady-state operation of a single-ion-species plasma, possibly with some external forcing $\mathcal{F}_{i\backslash e}$ and inertial forces $F_{ci||}$:

$$-n_i F_{ci||} = Z e n_i E_{||} - \nabla_{||} p_i + m_i n_i \nu_{ie}(v_{e||} - v_{i||}) + n_i \mathcal{F}_i \tag{20}$$

and

$$0 = -e n_e E_{||} - \nabla_{||} p_e + m_e n_e \nu_{ei}(v_{i||} - v_{e||}) + n_e \mathcal{F}_e. \tag{21}$$

Here $Z$ is the ion charge state, $e$ is the elementary charge, $p_s$ is the pressure of species $s$, $m_s$ is the mass of species $s$, and $\nu_{ss'}$ is the momentum transfer frequency for species $s$ and $s'$. The parallel subscript denotes the component parallel to $\mathbf{B}$ – for example, $E_{||} = \mathbf{E} \cdot \mathbf{B}/B$. Suppose $T_i$ and $T_e$ are constant and $n_e = Z n_i$.

Define

$$\xi_+ \doteq n_i \mathcal{F}_i + n_e \mathcal{F}_e \tag{22}$$

$$\xi_- \doteq n_i \mathcal{F}_i - n_e \mathcal{F}_e. \tag{23}$$

Then

$$\nabla_{||}(p_i + p_e) = n_i F_{ci||} + \xi_+ \tag{24}$$

and

$$\eta j_{||} = E_{||} + \frac{-\nabla_{||}(p_i - p_e) + n_i F_{ci||} + \xi_-}{2 Z e n_i}. \tag{25}$$

Here $\eta \doteq m_e \nu_{ei}/e^2 n_e$. Eq. (25) can be rewritten as

$$\eta j_{||} = E_{||} + \frac{T_e}{Z T_e + T_i} \frac{F_{ci||}}{e} + \frac{1}{2 e n_e}\left(\frac{Z T_e - T_i}{Z T_e + T_i} \xi_+ + \xi_-\right). \tag{26}$$

In the absence of momentum injection, $\xi_+ = \xi_- = 0$, and the current is proportional to the deviation of the electric field from its "natural" ambipolar value. The same effect appears in the case of more than one ion species. However, it is analytically much more complicated to describe due to the proliferation of additional simultaneous equations as more species are included.

Eq. (26) suggests that there are two strategies with which it might be possible to maintain a parallel electric field. The first is to use external forcing (noninductive current drive) to maintain some $E_{||}$, paying whatever energetic cost is associated with the relaxation of the plasma. The second is to adjust the ambipolar field to which the parallel conductivity pushes $E_{||}$. The first allows for a wider range of outcomes, but

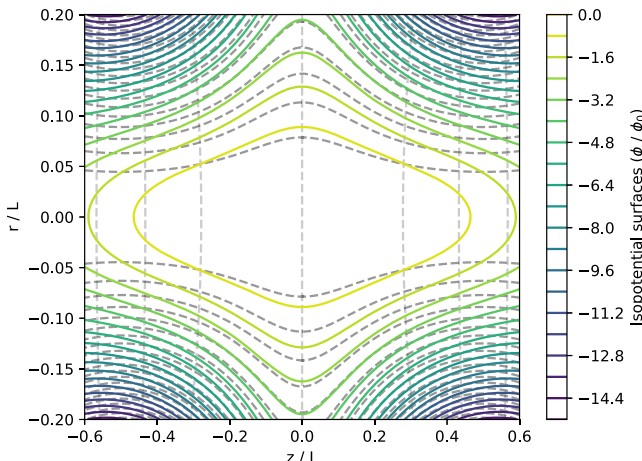

**Fig. 1 | Example of closed isopotential surfaces.** The colored curves show the isopotential surfaces for the example potential discussed in **An example**, such that the $\mathbf{E} \times \mathbf{B}$ rotation frequency does not vary along field lines. The horizontally- and vertically-oriented dotted curves trace out the level contours for $\psi$ and $\chi$, respectively.

the second avoids the problem of very large energy costs when $\eta$ is small. The remainder of this paper will focus on the latter strategy.

There is neither $j_\parallel E_\parallel$ Ohmic dissipation nor any need for external forcing when

$$\phi(\chi,\psi) - \phi(0,\psi) = \frac{T_e}{ZT_e + T_i} \frac{m_i}{2e} \left[ \Omega_E(\chi,\psi)^2 r^2 - \Omega_E(0,\psi)^2 r_0^2 \right], \quad (27)$$

where $r_0$ is the value of $r$ when $\chi = 0$ for a given flux surface $\psi$. This expression follows from integrating Eq. (26), and the $\mathbf{E} \times \mathbf{B}$ rotation has been taken to be dominant. Expressions closely related to Eq. (27) have long been known in the literature; this parallel variation in $\phi$ is sometimes called the ambipolar potential[43–46]. Eq. (27) can be written equivalently as

$$\phi(\chi,\psi) - \phi(0,\psi) = \frac{T_e}{ZT_e + T_i} \frac{m_i}{2e} \left[ \left( \frac{\partial \phi}{\partial \psi} - \gamma \frac{\partial \phi}{\partial \chi} \right)^2 \Big|_{(\chi,\psi)} r^2 - \left( \frac{\partial \phi}{\partial \psi} - \gamma \frac{\partial \phi}{\partial \chi} \right)^2 \Big|_{(0,\psi)} r_0^2 \right]. \quad (28)$$

There are a few things to point out about Eq. (28). First: this condition can also be derived by enforcing that the particles are Gibbs-distributed along field lines (though not necessarily across field lines). This makes sense; if the distributions are Gibbs-distributed in the parallel direction, then we should expect parallel currents to vanish. Second: if species $s$ is Gibbs-distributed along field lines (and if the plasma is isothermal) then we also have that $\varphi_s = \varphi_{s,\perp}(\psi)$; the electrochemical potential is a flux function, and each flux surface will isorotate. This means that potentials satisfying Eq. (28) avoid not only the dissipation associated with parallel currents but also the dissipation associated with shear along flux surfaces. Note, however, that in cases where the centrifugal force is not the gradient of a potential, the ions may not have a well-defined Gibbs distribution. In these cases, $j_\parallel = 0$ leads to isorotation of the electrons, but the ions may still have some shear. However, this shear is suppressed when either (1) the rotation frequency is small compared with the ion gyrofrequency or (2) the centrifugal force is close to being the gradient of a scalar function.

## Challenges

Solutions to Eq. (28) have desirable properties, but they come with significant challenges if they are to lead to closed isopotential

surfaces. The first of these has to do with the magnitude of the variation of $\phi$ in the parallel and perpendicular directions. It is clearest to see in the case where $\phi$ can be decomposed so that $\phi = \phi_\parallel(\chi) + \phi_\perp(\psi)$ and where $\mathbf{B}$ is a vacuum field. In this case, Eq. (28) becomes

$$\Delta\phi_\parallel = -\frac{T_e}{ZT_e + T_i} \frac{m_i}{2e} \Omega_E^2 (r_0^2 - r^2), \quad (29)$$

where $\Delta\phi_\parallel \doteq \phi_\parallel(\chi) - \phi_\parallel(0)$. If $E_\perp = -\Delta\phi_\perp / L_\perp$ for some perpendicular length scale $L_\perp$, then this can be rewritten as

$$\frac{\Delta\phi_\parallel}{\Delta\phi_\perp} = \frac{1}{2}\left(\frac{ZT_e}{ZT_e + T_i}\right) \frac{\Omega_E}{\Omega_{ci}} \frac{r_0^2 - r^2}{rL_\perp}. \quad (30)$$

Here $\Omega_{ci} \doteq ZeB/m_i$ and we have taken $\Omega_E = \Delta\phi_\perp / rL_\perp B$. The Brillouin limit requires that $\Omega_E/\Omega_{ci} < 1/4$; beyond this limit (which depends on the sign of the electromagnetic fields), the plasma cannot be confined. Then, assuming $E_\perp > 0$, $\Delta\phi_\parallel = \phi_\perp$ is only realizable if

$$L_\perp < \frac{1}{8} \frac{r_0^2 - r^2}{r}. \quad (31)$$

This suggests that in a cylindrically symmetric system, the plasma must occupy only a thin annular region (such that the perpendicular length scale can be small compared with the radius).

This constraint can be seen from a different perspective by rewriting Eq. (30) as

$$\frac{\Delta\phi_\parallel}{\Delta\phi_\perp} = \frac{1}{2}\left(\frac{ZT_e}{ZT_e + T_i}\right) \mathrm{Ma}_0 \frac{\rho_{Li}}{L_\perp} \frac{r_0^2 - r^2}{rr_0}. \quad (32)$$

Here $\rho_{Li}$ is the ion Larmor radius and $\mathrm{Ma}_0$ is the ratio of $v_{E\times B}$ and the ion thermal velocity, evaluated at $r_0$. If $B \propto r^{-2}$, then $\rho_{Li} \propto r^2$. Moreover, if at a given $z$ the plasma occupies a thin range of radii,

$$\psi(r + \delta r, z) - \psi(r,z) = \int_0^{r+\delta r} r' B_z \, dr' - \int_0^r r' B_z dr \quad (33)$$

$$= rB_z(r) \, \delta r + \mathcal{O}\left(\frac{\delta r^2}{r^2}\right), \quad (34)$$

so for a thin annular geometry, we should roughly expect $L_\perp \propto r$. Then $\rho_{Li}/L_\perp \propto r$. Using this,

$$\frac{\Delta\phi_\parallel}{\Delta\phi_\perp} = \frac{1}{2}\left(\frac{ZT_e}{ZT_e + T_i}\right) \mathrm{Ma}_0 \left(\frac{\rho_{Li}}{L_\perp}\right)_{r_0} \frac{r_0^2 - r^2}{r_0^2}. \quad (35)$$

In order for a configuration to have good cross-field particle confinement times, the width of the plasma likely needs to span several Larmor radii at least. Eq. (35) suggests that this constraint can be satisfied only when the Mach number is relatively large.

A related constraint suggests that fully freestanding rotation would require not only a large Mach number, but a very large parallel voltage drop. If Ma is the ratio of $v_{E\times B}$ and the ion thermal velocity evaluated at $r$, then using the definition of $L_\perp$ from the beginning of this section,

$$\mathrm{Ma} = \frac{Ze\Delta\phi_\perp}{T_i} \frac{\rho_{Li}}{L_\perp}. \quad (36)$$

That is, the Mach number is approximately the perpendicular drop in electrostatic potential energy (compared with the ion temperature) over one ion Larmor radius. For a configuration with supersonic rotation and a reasonable number of Larmor radii in width, the

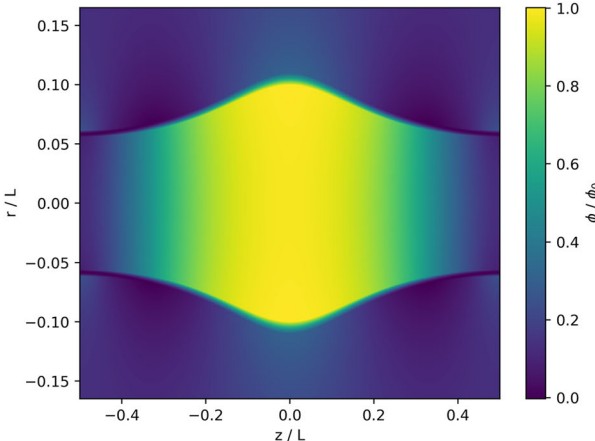
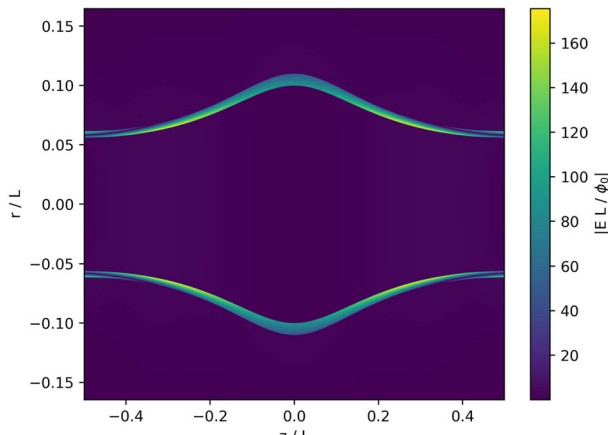

**Fig. 2 | Low-dissipation example equilibrium.** Example solution from Eq. (37), with total ($\mathbf{E} \times \mathbf{B}$ and diamagnetic) flux surface isorotation and vanishing parallel current. The left-hand panel shows the potential $\phi$ and the right-hand panel shows $|E|$. This particular solution has the nice property that there are regions near the edge with very small electric fields, despite supporting (potentially large) fields in the interior. The scripts used to produce these plots can be found in ref. 64.

perpendicular drop in electrostatic potential energy must be large compared with $T_i$. This means that whenever $\Delta\phi_\parallel \sim \Delta\phi_\perp$, the parallel drop must also be large compared with $T_i$.

In the existing literature on rotating plasmas, it is common to assume that the parallel variation in $\phi$ is ordered to be very small compared with the cross-field variation[44–48]. One way of understanding the challenges described in this section is that closing the isopotential surfaces requires finding a way to break that ordering. In particular, note that $\Delta\phi_\parallel \sim \Delta\phi_\perp$ tends to require very fast (often supersonic) diamagnetic flows since the pressure forces cannot be ordered small compared with $e\mathbf{E}$.

Nonetheless, in principle we can conclude that it is possible to maintain very high voltage drops across a plasma while incurring little dissipation. However, it is worthwhile to keep in mind what it would mean for the particles to be Gibbs-distributed along field lines if the potential drops were very large. A megavolt-scale potential drop across a plasma with a temperature on the scale of keV would require many $e$-foldings of density dropoff along each field line, and would lead to equilibria that require densities low enough to be challenging to realize in a laboratory.

## Example solution

Low-dissipation solutions of the kind described by Eq. (28) are not always straightforward to find. However, it is possible to find solutions to Eq. (28) that are valid for any choice of (cylindrically symmetric) field. For example,

$$\phi(\chi,\psi) = \left( \phi(\chi,\psi_i)^{1/2} \pm \frac{1}{2} \sqrt{\frac{2e}{m_i}\frac{ZT_e + T_i}{T_i}} \int_{\psi_i}^{\psi} \frac{\mathrm{d}\psi'}{r(\chi,\psi')} \right)^2 \quad (37)$$

solves Eq. (28) for any choice of nonnegative $\phi(\chi, \psi_i)$. The magnetic field geometry appears through the coordinate transformation $r(\chi, \psi)$ (the radial coordinate expressed in terms of $\chi$ and $\psi$). Depending on the prefactor of the integral in Eq. (37), this family of solutions can lead to closed isopotential surfaces.

A solution of this form is plotted in Fig. 2. Plotting solutions of this kind requires choosing which region will be occupied by plasma, with $\phi$ governed by Eq. (37), and which will instead be the vacuum solution determined by Laplace's equation. The constraints described in **Challenges** suggest limiting the plasma to appear only within some relatively thin range of flux surfaces. For the particular example plotted in Fig. 2, the plasma is restricted to occupy the region in which $\psi_i < \psi < \psi_f$, where $\psi_i = 0.00237 B_0 L^2$ and $\psi_f = 0.00284 B_0 L^2$

(corresponding to $0.1 < r/L < .11$ at the midplane). This example uses $\phi(\chi,\psi_i) = \phi_0 e^{-L^2 \chi^2 / B_0^2}$. It uses the negative branch of Eq. (37) and $(eB_0^2 L^2 / 2m_i\phi_0)[(ZT_e + T_i)/T_i]$ set to $10^4$. The underlying field is described by Eqs. (15) and (18).

These choices are arbitrary, so it is important to understand Fig. 2 as an illustrative example rather than the definitive embodiment of this class of solutions. It does have the interesting property that there is a region near the ends of the plasma where the electric field becomes small. This suggests that an endplate or plasma-facing component shaped in the right way could experience a much smaller electric field than the field present in the plasma interior.

## Initializing desired equilibria

What might be needed in order to drive an equilibrium like the one described above in a practical device? The basic field and device geometry would not be so different from a conventional rotating-mirror experiment; with good enough cross-field confinement, one could imagine initializing an annular density profile in a linear device. The more difficult problem is likely how to drive the necessary electric field structure.

Two promising techniques for driving and controlling electric fields in the interior of plasma are RF current drive and neutral beams[14–19]. In either case, the idea is to impose some torque on the interior of the plasma; there is a one-to-one mapping between the local torque and the cross-field current drive. The ability of these techniques to produce a particular potential profile relies on their ability to deposit angular momentum precisely in the desired locations in the plasma. In the case of RF waves, for example, this depends on finding a wave with a spatial damping profile that will put the angular momentum where it needs to go in order to produce the desired $\phi(\chi, \psi)$.

Note that even for a particular choice of $\phi(\chi,\psi)$, the wave and neutral-beam deposition profiles would depend on additional free parameters such as the plasma density. For any given parameter regime and choice of equilibrium, attaining a particular equilibrium will require an array of RF antennas, neutral beam launchers, or some combination of the two.

## Stability of desired equilibria

Virtually all plasma confinement devices are subject to instabilities of one kind or another. Now, it is important to keep in mind that the equilibria proposed here constitute a broad class of configurations, and the instabilities that will be most important for one equilibrium in

that class may be different from those that are most important for another. Still, it is worth pointing out which instabilities are likely to be of greatest concern.

These equilibria fall within the broader category of rotating-mirror configurations, so many of the instabilities to consider are the same ones that challenge all devices in this class. These include magnetohydrodynamic (MHD) flute modes as well as mirror micro-instabilities (particularly loss-cone modes)[25].

Rotating mirrors generally have stability advantages over their non-rotating counterparts for two reasons. One of these has to do with sheared rotation – that is, not the shear along flux surfaces that the equilibria in this paper avoid by construction, but the shear between flux surfaces. There is evidence that shear flow can suppress MHD modes in rotating mirrors[6,49], and more generally that sufficient shear can suppress turbulent transport[50,51]. The second reason for their improved stability is that centrifugal mirror traps tend to have more isotropic velocity-space distributions than do conventional mirrors, with sonic or supersonic rotation sufficient to suppress many of the major loss-cone instabilities[52–54]. None of this is to say that all equilibria satisfying Eq. (28) will necessarily avoid these instabilities. Rather, it suggests that the subset of these solutions with (1) sonic or supersonic rotation and (2) sufficient shear between flux surfaces may be able to avoid them. Moreover, there are situations in which rotation can make stabilization more difficult. For example, even though shear flow tends to stabilize flute modes, centrifugal forces tend to destabilize them, so the net effect of the rotating flow depends on the balance between these two effects[3].

The special properties of these particular rotating-mirror equilibria may also make some instabilities more challenging. If the plasma occupies a thin annular volume, then solutions with higher peak densities must also have large density gradients. This is a source of free energy that can drive modes like the drift wave instabilities. There is a large literature on these modes and a variety of strategies to mitigate them, including cross-field shear and geometric strategies[55–57].

## Discussion

The conventional picture of an open-field-line $\mathbf{E} \times \mathbf{B}$ rotating plasma requires that each flux surface also be a surface of (approximately) constant voltage. This comes with certain constraints. Indeed, it is difficult to imagine operating such a device beyond some maximal voltage drop; even though the plasma itself can tolerate large fields without problem, the field lines in open configurations intersect with the solid material of the device, and material components cannot survive fields beyond some threshold. Van de Graaff-type devices can sustain voltages in the tens of megavolts[58]; it is very difficult to prevent material breakdown beyond this level (fully ionized plasma, of course, does not have this difficulty). If flux surfaces are surfaces of constant potential, then high voltages across the interior of the plasma necessarily result in high voltages across the material components, and this limits the interior voltage drop.

Limitations on the achievable electric fields are important in a variety of applications. In centrifugal traps, any limitation on the electric field can be understood as a limit on the maximum plasma temperature. To see this, note that the limit on the temperature that can be contained is set by the centrifugal potential, which is determined by the rotation velocity. This, in turn, depends on the electric and magnetic fields. Some advantages can be had by reducing the magnetic field strength (since the $\mathbf{E} \times \mathbf{B}$ velocity goes like $E/B$), but perpendicular particle confinement requires that the field not be reduced too much.

There are some applications for which open $\mathbf{E} \times \mathbf{B}$ configurations are feasible only if the voltage drops in the interior of the plasma can be very high. For example, thermonuclear devices burning aneutronic

fuels are likely to require very high temperatures. Limitations on the achievable electric fields could determine whether or not centrifugal traps are viable for such applications. This paper considers what might be required in order to relax these limitations.

First, it is important to avoid shear of the angular velocity along flux surfaces (that is, to maintain isorotation). It is well known that isorotation of the $\mathbf{E} \times \mathbf{B}$ rotation frequency follows any time the flux surfaces are isopotentials, but we show here that the general conditions for isorotation are much less strict than that.

Second, it is important to avoid excessive Ohmic dissipation from parallel electric fields. Some plasmas have higher parallel conductivities than others, but especially for hot plasmas, the conductivity (and the associated dissipation) can be very high. Fortunately, in a rotating plasma, the parallel currents are not proportional to the parallel fields. If the parallel fields are close to the "ambipolar" fields, the Joule heating vanishes. The ambipolar fields have the nice property that they also produce isorotation of the combined $\mathbf{E} \times \mathbf{B}$ and diamagnetic flows.

Eliminating these sources of dissipation would not result in a perfectly dissipationless system, even if all instabilities can be suppressed. Cross-field transport – at least at the classical level – would still lead to some losses (as is the case in any magnetic trap), as would cross-field viscosity. However, these effects are suppressed at high magnetic fields[59–61], so the elimination of these sources could lead to a configuration that is at least as long-lived as the timescale of Braginskii's cross-field viscosity[34], which is typically many orders of magnitude longer than the parallel Ohmic dissipation time.

In many cases, the parallel ambipolar fields are small compared with the perpendicular fields driving the rotation. In order for a configuration to have a large voltage drop in the plasma interior and a small voltage drop at the edges of the device, the parallel and perpendicular voltage drops must be comparable. We show in Challenges that this is challenging but possible. It requires supersonic rotation, and it requires a configuration for which the perpendicular length scale is small compared with the total radius (e.g., a relatively thin annulus of plasma in a larger cylindrical device). In principle, this opens up the possibility of a much wider design space for open-field-line rotating devices than has previously been considered. Note that these solutions not only do not require end-electrode biasing but that they cannot be produced by end-electrodes alone. That is, actually setting up fields of this kind is likely to require electrodeless techniques for driving voltage drops, whether that be wave-driven, neutral beams, or something else.

Note that the strategy discussed here results in large rotation in a simple mirror geometry; there remains the opportunity to sequence multiple such rotating mirrors much in the same way that has been approached for simple non-rotating mirrors[62]. Also, note that the strategy described in this paper is not the only possible way to reduce the fields across the material boundary of a plasma device. It is also possible to reduce these fields geometrically. If the potential is constant along every field line, and if every field line impinges somewhere on the material components of the device, then the total voltage drop between the highest and the lowest point is fixed. However, the field can be reduced by arranging for the field lines to expand over a larger region before they impinge on the surface so that the local fields are reduced (not entirely unlike a diverter[63]). This strategy is shown, in cartoon form, in Fig. 3. However, it has clear limitations; in a cylindrically symmetric system, doubling the radius of the outer vessel reduces the fields by a factor of two. Similarly, some advantage can be gained by manipulating the angle of incidence of the magnetic field on the plasma-facing components, but this can only be pushed so far. Very large field reductions would require correspondingly large geometric expansions, and may not always be a practical alternative to the solution discussed here.

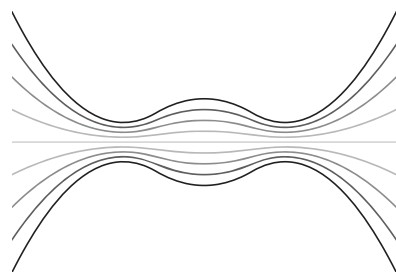

**Fig. 3 | Geometric strategy for reducing edge electric fields.** This cartoon shows an alternative strategy for reducing the electric field across material components: to construct a region outside of the main mirror cell with expanding magnetic fields so that the physical distance between field lines impinging on the outer vessel boundary is larger. If field lines are isopotentials, then this reduces the resulting fields at the boundaries by a factor that scales with the field-line spacing.

## Methods
### Vacuum solutions for the potential
If the plasma occupies some region $\psi_i \leq \psi \leq \psi_f$, and we specify $\phi$ within this region, we may still wish to compute the isopotential contours for $\psi < \psi_i$ and $\psi > \psi_f$. If there is no free charge in the unoccupied regions, $\phi$ must satisfy Laplace's equation in these areas:

$$\nabla^2 \phi = 0. \tag{38}$$

Assuming cylindrical symmetry, this is

$$\frac{1}{r}\frac{\partial}{\partial r}\left(r\frac{\partial \phi}{\partial r}\right) + \frac{\partial^2 \phi}{\partial z^2} = 0. \tag{39}$$

For solutions that are periodic in $z$, with boundary conditions such that $\phi$ vanishes at $z = \pm L/2$, $\phi(\psi < \psi_i)$ can be written as the series solution

$$\phi = \sum_{n=0}^{\infty} A_n \cos\left(\frac{2\pi n z}{L}\right) I_0\left(\frac{2\pi n r}{L}\right) \tag{40}$$

and $\phi(\psi > \psi_f)$ can be written as

$$\phi = C_0 + \sum_{n=1}^{\infty} C_n \cos\left(\frac{2\pi n z}{L}\right) K_0\left(\frac{2\pi n r}{L}\right). \tag{41}$$

Here $I_0$ is a modified Bessel function of the first kind, $K_0$ is a modified Bessel function of the second kind, and the $A_n$ and $C_n$ are scalar coefficients. This choice of eigenfunctions imposes the constraint that $\phi$ must be well-behaved near $r = 0$ for the inner solution and must converge to some constant value when $r \to \infty$ for the outer solution. In the context of this problem, the $A_n$ and $C_n$ are chosen to match the boundary curves $\phi(\chi, \psi_i)$ and $\phi(\chi, \psi_f)$, respectively. For the particular case shown in Fig. 2, the first ten terms each of the $A_n$ and $C_n$ are used to fit the boundary.

### On twisting fields
We sometimes consider systems in which the $\mathbf{E} \times \mathbf{B}$ flow is axially sheared; that is, if $\mathbf{v}_{E \times B} = r\Omega_E \hat{\theta}$, $\partial \Omega_E / \partial \chi \neq 0$. If $\mathbf{E} = -\nabla \phi$, this can result if $\partial \phi / \partial \chi \neq 0$.

Our intuition from the ideal MHD is that this ought to lead the field lines to twist up. The ideal MHD induction equation states that

$$\frac{\partial \mathbf{B}}{\partial t} = \nabla \times (\mathbf{v} \times \mathbf{B}), \tag{42}$$

so that

$$\frac{\partial \mathbf{B}}{\partial t} = \nabla \times \left[r^2 \Omega_E \nabla \theta \times \mathbf{B}\right] \tag{43}$$

$$= \nabla \times (\Omega_E \nabla \psi) \tag{44}$$

$$= \nabla \Omega_E \times \nabla \psi. \tag{45}$$

This would imply that

$$\frac{\partial B_\theta}{\partial t} = 0 \quad \text{iff.} \quad \frac{\partial \Omega_E}{\partial \chi} = 0. \tag{46}$$

In other words, the ideal MHD induction equation appears to suggest that the axial shear of $\Omega_E$ twists up field lines.

However, this is not the case. To see why, note that this argument (and all of the intuition behind it) relies on mixing the ideal MHD induction equation with an $\mathbf{E} \times \mathbf{B}$ drift that is not consistent with the ideal MHD. In an ideal MHD,

$$\mathbf{E} = -\mathbf{v} \times \mathbf{B}; \tag{47}$$

the theory does not permit any component of $\mathbf{E}$ in the direction of $\mathbf{B}$. (In rotating mirrors, we get a parallel component of $\phi$ by including electron-pressure corrections in an extended-MHD Ohm's law, but this is not essential to the argument).

Consider instead the original form of Faraday's equation:

$$\frac{\partial \mathbf{B}}{\partial t} = -\nabla \times \mathbf{E}. \tag{48}$$

If $\mathbf{E} = -\nabla \phi$, we do not get twisting of the field lines, no matter what kind of dependences $\phi$ might have. So, in a rotating mirror, it is incorrect to conclude that non-isorotation must necessarily lead to $B_\theta$.

If we derive the form of the steady-state $\phi$ that results from, e.g., electron pressure, we find that the corresponding correction term to the MHD induction equation always cancels any field-line twisting – as we know that it must, from Faraday's equation.

## Code availability
The plotting scripts used to make the numerical figures can be found on Zenodo at https://zenodo.org/doi/10.5281/zenodo.10621240, or on GitHub at https://github.com/ekolmes/voltageDropPlots.

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

## Acknowledgements
The authors thank Alex Glasser, Mikhail Mlodik, and Tal Rubin for helpful conversations. E. J. K., I. E. O., J.-M. R. and N. J. F. acknowledge support from ARPA-E Grant No. DE-AR0001554. E. J. K. and I. E. O. also received support from the DOE Fusion Energy Sciences Postdoctoral Research Program, administered by the Oak Ridge Institute for Science and Education (ORISE) and managed by Oak Ridge Associated Universities (ORAU) under DOE Contract No. DE-SC0014664.

## Author contributions
E. J. K., I. E. O., J.-M. R. and N. J. F. all participated in the analysis presented here, and in the preparation of the manuscript.

## Competing interests
The authors declare no competing interests.
