## [Peer Review File · Nature Communications]

Massive, Long-Lived Electrostatic Potentials in a Rotating Mirror PlasmaReviewer #1 (Remarks to the Author):

Please see the attached pdf.

Review – Kolmes et al, Massive, Long-Lived Electrostatic Potentials in a Rotating Mirror Plasma

This article describes an interesting variation on a common plasma configuration, in which large potential drops can be sustained in the core of a plasma (such as in centrifugal fusion devices) while overcoming the limitations normally set by the impingement of field lines on material boundaries. The authors describe how the potential drop in the plasma can be created and maintained using a first principles analysis of an axisymmetric plasma with $E \times B$ rotation. The authors consider how, in order to achieve plasma confinement and the maintenance of the desired fields within the core plasma, closed isopotential surfaces can be generated within the plasma while minimizing dissipation. This interesting configuration could be relevant to fusion.

The paper is well-written and effectively conveys the significance of the cases. The first principles analysis is interesting and accurate for the cases considered. Although I have some questions, overall the authors have done a commendable job not only in their analysis, but also in their consideration of the physical implications of these analyses.

1. In part III, the authors describe conditions allowing parallel electric fields to be maintained and show the relationship established between parallel and perpendicular potentials, which results in a characteristic length scale for the plasma being established (Equation 30). Essentially, the dimensions of the plasma must be strongly restricted to achieve the necessary condition for the parallel electric field. Could the authors elaborate on how this would be practically achievable? The criteria established appear to be sound, but I'm not aware of any devices where this condition could practically be met?
2. Can the authors discuss dissipation expected to arise from the instabilities created in the configuration investigated? The strong diamagnetic and $E \times B$ drifts, in addition to the steep density gradients this configuration would require in order to sustain the large fields within the plasma, would create ideal conditions for instabilities which would ultimately limit the confinement. The idealized analysis does not account for such effects, which would be very significant.
3. Can the authors elaborate on the significance of their description of the potentials as "long-lived" – I just want to be certain I'm understanding the full import of their arguments. Are they merely referring to the fact that the (large) fields can be sustained in the configurations they describe, without intending to describe the long-term stability of such fields?
4. I wonder if the authors can comment on what the effects would be if they considered a multispecies plasma – would this affect the ability to sustain the fields and confinement?

Reviewer #2 (Remarks to the Author):

This manuscript presents a new family of configurations for a magnetized plasma device of the mirror type, intended to sustain very high voltage drops across the field lines. The concept is interesting, and the derived sufficient condition for isorotation is in itself of general interest. Thus, in my view the paper deserves publication, after a review according to the following points:

- 1) On page 2 it is said that the limit where gamma tends to zero is the low-beta (vacuum field) limit. First of all, it should be pointed out that the low-beta limit is not the same as the vacuum field limit, because in the former case a current can still be present within the plasma. The authors should clarify this issue, and also better explain why such limit is the one where gamma tends to zero, as this may not be apparent to the reader.
- 2) The problem of how to achieve in practice the configurations described in the paper is treated only in very generic terms. The authors should make an effort to be more precise, that is, without entering into technical details, give a tentative design of how such device could be constructed, especially as far as obtaining the correct electric field configuration is concerned.
- 3) When realizing one of such configurations in practice, experience from different fields of experimental plasma physics suggests that the neat properties described in the paper may be spoiled by the onset of instabilities and ensuing turbulence, giving rise to enhanced cross-field transport. Of course, the idealized treatment given in the paper does not lose its interest because of this, but the authors should at least mention this both in their introduction and in the conclusions, possibly giving their feelings about the impact of such phenomena on their proposed configuration, based on existing experience in mirror devices, but taking into account that strong electric fields can be an additional source of free energy for instability development.

List of Changes and Reply to Reviewer Reports

We thank both of the Reviewers for their time, thorough reading, and helpful comments. We very much appreciate it. We have made a variety of changes to the manuscript in order to address their concerns and suggestions (as well as some formatting changes per the instructions of the editorial office). We will respond to the Reviewers' comments and enumerate our resulting changes point-by-point below.

Reply to Report of Reviewer 1

Report 1, Part 1

This article describes an interesting variation on a common plasma configuration, in which large potential drops can be sustained in the core of a plasma (such as in centrifugal fusion devices) while overcoming the limitations normally set by the impingement of field lines on material boundaries. The authors describe how the potential drop in the plasma can be created and maintained using a first principles analysis of an axisymmetric plasma with $E \times B$ rotation. The authors consider how, in order to achieve plasma confinement and the maintenance of the desired fields within the core plasma, closed isopotential surfaces can be generated within the plasma while minimizing dissipation. This interesting configuration could be relevant to fusion.

The paper is well-written and effectively conveys the significance of the cases. The first principles analysis is interesting and accurate for the cases considered. Although I have some questions, overall the authors have done a commendable job not only in their analysis, but also in their consideration of the physical implications of these analyses.

1. In part III, the authors describe conditions allowing parallel electric fields to be maintained and show the relationship established between parallel and perpendicular potentials, which results in a characteristic length scale for the plasma being established (Equation 30). Essentially, the dimensions of the plasma must be strongly restricted to achieve the necessary condition for the parallel electric field. Could the authors elaborate on how this would be practically achievable? The criteria established appear to be sound, but I'm not aware of any devices where this condition could practically be met?

Reply

We thank the Reviewer for their time and comments. In response to this question (and to a closely related question asked by the other Reviewer), we have added a subsection entitled “Initializing Desired Equilibria,” which reads as follows:

What might be needed in order to drive an equilibrium like the one described above in a practical device? The basic field and device geometry would not be so different from a conventional rotating-

mirror experiment; with good enough cross-field confinement, one could imagine initializing an annular density profile in a linear device. The more difficult problem is likely how to drive the necessary electric field structure.

Two promising techniques for driving and controlling electric fields in the interior of a plasma are RF current drive and neutral beams [14-19]. In either case, the idea is to impose some torque on the interior of the plasma; there is a one-to-one mapping between the local torque and the cross-field current drive. The ability of these techniques to produce a particular potential profile relies on their ability to deposit angular momentum precisely in the desired locations in the plasma. In the case of RF waves, for example, this depends on finding a wave with a spatial damping profile that will put the angular momentum where it needs to go in order to produce the desired $\phi(\chi, \psi)$.

Report 1, Part 2

Can the authors discuss dissipation expected to arise from the instabilities created in the configuration investigated? The strong diamagnetic and $E \times B$ drifts, in addition to the steep density gradients this configuration would require in order to sustain the large fields within the plasma, would create ideal conditions for instabilities which would ultimately limit the confinement. The idealized analysis does not account for such effects, which would be very significant.

Reply

We agree that this is an important issue to discuss. We have added a subsection entitled “Stability of Desired Equilibria” immediately before the Discussion, which reads as follows:

Virtually all plasma confinement devices are subject to instabilities of one kind or another. Now, it is important to keep in mind that the equilibria proposed here constitute a broad class of configurations, and the instabilities that will be most important for one equilibrium in that class may be different from those that are most important for another. Still, it is worth pointing out which instabilities are likely to be of greatest concern.

These equilibria fall within the broader category of rotating-mirror configurations, so many of the instabilities to consider are the same ones that challenge all devices in this class. These include magneto-hydrodynamic (MHD) flute modes as well as mirror microinstabilities (particularly loss-cone modes) [25].

Rotating mirrors generally have stability advantages over their non-rotating counterparts for two reasons. One of these has to do with sheared rotation – that is, not the shear along flux surfaces that the equilibria in this paper avoid by construction, but the shear between flux surfaces. There is evidence that shear flow can suppress MHD modes in rotating mirrors [6, 50], and more generally that sufficient shear can suppress turbulent transport [51, 52]. The second reason for their improved stability is that centrifugal mirror traps tend to have more isotropic velocity-space distributions than do conventional mirrors, with sonic or supersonic rotation sufficient to suppress many of the major loss-cone instabilities [53-55]. None of this is to say that all equilibria satisfying Eq. (27) will necessarily avoid these instabilities. Rather, it suggests that the subset of these solutions with (1) sonic or supersonic rotation and (2) sufficient shear between flux surfaces may be able to avoid them. Moreover, there are situations in which rotation can make stabilization more difficult. For example, even though shear flow tends to stabilize flute modes, centrifugal forces tend to destabilize them, so the net effect of the rotating flow depends on the balance between these two effects [3].

The special properties of these particular rotating-mirror equilibria may also make some instabilities more challenging. If the plasma occupies a thin annular volume, then solutions with higher peak densities must also have large density gradients. This is a source of free energy that can drive modes like the drift wave instabilities. There is a large literature on these modes and a variety of strategies to mitigate them, including cross-field shear and geometric strategies [56-58].

We also added the caveat that the contained voltage drop needed to be stable as well as attainable to the first sentence of the last paragraph of the introduc-

tion, and the caveat that the classical cross-field dissipation discussed in the sixth paragraph of the Discussion presupposes that the instabilities can be suppressed.

Report 1, Part 3

Can the authors elaborate on the significance of their description of the potentials as “long-lived” – I just want to be certain I’m understanding the full import of their arguments. Are they merely referring to the fact that the (large) fields can be sustained in the configurations they describe, without intending to describe the long-term stability of such fields?

Reply

We intended “long-lived” to be a statement about dissipation – that it is possible for these fields to be maintained for a long period of time without relaxing due to power loss from conductivity or viscosity. In retrospect, this should have been mentioned explicitly somewhere in the paper. To correct this, we have changed the last sentence in the third-to-last paragraph of the conclusion from

However, these effects are suppressed at high magnetic fields [51-53].

to

However, these effects are suppressed at high magnetic fields [51-53], so the elimination of these sources could lead to a configuration that is at least as long-lived as the timescale of Braginskii’s cross-field viscosity [34], which is typically many orders of magnitude longer than the parallel Ohmic dissipation time.

Report 1, Part 4

I wonder if the authors can comment on what the effects would be if they considered a multispecies plasma – would this affect the ability to sustain the fields and confinement?

Reply

This is something that we have spent some time looking into. The short answer is that the key effects described in the paper work in much the same way if we

allow for several ion species. The longer answer – and the reason why we didn't address this in the original version of the manuscript – is that the introduction of additional ion species keeps the basic behavior the same but makes the analytic descriptions of the density profiles much more complicated.

To see why, consider the simpler (but closely related) scenario of a plasma sitting in a constant gravitational field with strength g . With a single ion species i , the quasineutral isothermal equilibrium is the one in which

$$n_e = Z_i n_0 e^{e\varphi/T_e} \quad (1)$$

$$n_i = n_0 e^{-(Z_i e\varphi + m_i g z)/T_i} \quad (2)$$

and the electrostatic potential φ must be

$$e\varphi = -\left(\frac{T_e}{T_i + Z_i T_e}\right) m_i g z. \quad (3)$$

This is the analog of the result found in the paper for a centrifugal potential.

When there are two ion species a and b , the corresponding problem is to find $\varphi(z)$ that satisfies the following:

$$\begin{aligned} & (Z_a n_{a0} + Z_b n_{b0}) \exp\left[\frac{e\varphi}{T_e}\right] \\ &= Z_a n_{a0} \exp\left[-\frac{Z_a e\varphi + m_a g z}{T_a}\right] + Z_b n_{b0} \exp\left[-\frac{Z_b e\varphi + m_b g z}{T_b}\right] \end{aligned} \quad (4)$$

for some n_{a0} and n_{b0} .

A solution to this implicit equation for φ will exist for each z , but it does not take the simple form that we had in the single-ion-species case.

The special profiles described in the paper are essentially those in which each species is Gibbs-distributed in the direction parallel to the magnetic field, with the centrifugal force playing the role of gravity and the electrostatic potential φ arranging itself to produce quasineutrality in much the same way.

With all of that in mind, we added a brief comment to the corresponding calculation in the paper, noting that

The same effect appears in the case with more than one ion species. However, it is analytically much more complicated to describe due

to the proliferation of additional simultaneous equations as more species are included.

We have avoided getting into this issue in much more detail in the text out of concern that too much additional information might make the narrative more difficult to follow.

Reply to Report of Reviewer 2

Report 2, Part 1

This manuscript presents a new family of configurations for a magnetized plasma device of the mirror type, intended to sustain very high voltage drops across the field lines. The concept is interesting, and the derived sufficient condition for isorotation is in itself of general interest. Thus, in my view the paper deserves publication, after a review according to the following points:

1) On page 2 it is said that the limit where gamma tends to zero is the low-beta (vacuum field) limit. First of all, it should be pointed out that the low-beta limit is not the same as the vacuum field limit, because in the former case a current can still be present within the plasma. The authors should clarify this issue, and also better explain why such limit is the one where gamma tends to zero, as this may not be apparent to the reader.

Reply

We thank the Reviewer for their time and comments. The distinction between low β and vacuum fields is a good point. We had in mind the scaling of the diamagnetic currents with the plasma pressure, but there are other mechanisms that could produce \mathbf{j} even when β is small. We have changed the language about the “low- β ” case to instead refer to a vacuum-field limit. We have also added the following explanatory text immediately after we make the point about $\gamma \rightarrow 0$:

This is possible because, in the absence of plasma currents, the curl of \mathbf{B} vanishes everywhere in the interior of the plasma and the Helmholtz decomposition can be written in terms of a pure scalar potential.

Report 2, Part 2

The problem of how to achieve in practice the configurations described in the paper is treated only in very generic terms. The authors should make an effort to be more precise, that is, without entering into technical details, give a tentative design of how such device could be constructed, especially as far as obtaining the correct electric field configuration is concerned.

Reply

We have added a new subsection to address this point (as well as a closely related point made by the other Reviewer). It is called “Initializing Desired Equilibria,” and it reads as follows:

What might be needed in order to drive an equilibrium like the one described above in a practical device? The basic field and device geometry would not be so different from a conventional rotating-mirror experiment; with good enough cross-field confinement, one could imagine initializing an annular density profile in a linear device. The more difficult problem is likely how to drive the necessary electric field structure.

Two promising techniques for driving and controlling electric fields in the interior of a plasma are RF current drive and neutral beams [14-19]. In either case, the idea is to impose some torque on the interior of the plasma; there is a one-to-one mapping between the local torque and the cross-field current drive. The ability of these techniques to produce a particular potential profile relies on their ability to deposit angular momentum precisely in the desired locations in the plasma. In the case of RF waves, for example, this depends on finding a wave with a spatial damping profile that will put the angular momentum where it needs to go in order to produce the desired $\phi(\chi, \psi)$.

Report 2, Part 3

When realizing one of such configurations in practice, experience from different fields of experimental plasma physics suggests that the neat properties described in the paper may be spoiled by the onset of instabilities and ensuing turbulence, giving rise to enhanced cross-field transport. Of course, the idealized treatment given in the paper does not lose its interest because of this, but the authors should at least mention this both in their introduction and in the conclusions, possibly giving their feelings about the impact of such phenomena on their proposed configuration, based on existing experience in mirror devices, but taking into account that strong electric fields can be an additional source of free energy for instability development.

Reply

We agree that this is an important issue to discuss. To address this point (as well as in response to a closely related issue raised by the other Reviewer), we have added a subsection entitled “Stability of Desired Equilibria” immediately before the Discussion, which reads as follows:

Virtually all plasma confinement devices are subject to instabilities of one kind or another. Now, it is important to keep in mind that the equilibria proposed here constitute a broad class of configurations, and the instabilities that will be most important for one equilibrium in that class may be different from those that are most important for another. Still, it is worth pointing out which instabilities are likely to be of greatest concern.

These equilibria fall within the broader category of rotating-mirror configurations, so many of the instabilities to consider are the same ones that challenge all devices in this class. These include magneto-hydrodynamic (MHD) flute modes as well as mirror microinstabilities (particularly loss-cone modes) [25].

Rotating mirrors generally have stability advantages over their non-rotating counterparts for two reasons. One of these has to do with sheared rotation – that is, not the shear along flux surfaces that the equilibria in this paper avoid by construction, but the shear between flux surfaces. There is evidence that shear flow can suppress MHD modes in rotating mirrors [6, 50], and more generally that sufficient shear can suppress turbulent transport [51, 52]. The second reason for their improved stability is that centrifugal mirror traps tend to have more isotropic velocity-space distributions than do conventional mirrors, with sonic or supersonic rotation sufficient to suppress many of the major loss-cone instabilities [53-55]. None of this is to say that all equilibria satisfying Eq. (27) will necessarily avoid these instabilities. Rather, it suggests that the subset of these solutions with (1) sonic or supersonic rotation and (2) sufficient shear between flux surfaces may be able to avoid them. Moreover, there are situations in which rotation can make stabilization more difficult. For example, even though shear flow tends to stabilize flute

modes, centrifugal forces tend to destabilize them, so the net effect of the rotating flow depends on the balance between these two effects [3].

The special properties of these particular rotating-mirror equilibria may also make some instabilities more challenging. If the plasma occupies a thin annular volume, then solutions with higher peak densities must also have large density gradients. This is a source of free energy that can drive modes like the drift wave instabilities. There is a large literature on these modes and a variety of strategies to mitigate them, including cross-field shear and geometric strategies [56-58].

We also added the caveat that the contained voltage drop needed to be stable as well as attainable to the first sentence of the last paragraph of the introduction, and the caveat that the classical cross-field dissipation discussed in the sixth paragraph of the Discussion presupposes that the instabilities can be suppressed.

Reviewer #1 (Remarks to the Author):

The interest and originality of this manuscript are clear, as the authors have outlined and analyzed a new configuration with relevance to improved plasma confinement. The analysis is well-done and clearly-presented. I have noted that the revisions to the manuscript reflect the comments made by myself and the other reviewer, the most important of which queried how such configurations could be created, and the role of instabilities in sustaining such configurations. I think the ideas proposed in this work might trigger ideas for experimental devices which could be the subject of future studies.

Reviewer #2 (Remarks to the Author):

The authors have properly addressed all the remarks, and therefore the manuscript is now appropriate for publication.